# Investing in Urban Blue–Green Infrastructure—Assessing the Costs and Benefits of Stormwater Management in a Peri-Urban Catchment in Oslo, Norway

**Gert-Jan Wilbers [1,*], Karianne de Bruin [1], Isabel Seifert-Dähnn [2], Wiebe Lekkerkerk [3], Hong Li [4] and Monserrat Budding-Polo Ballinas [1]**

1   Wageningen Environmental Research, Wageningen University and Research, Droevendaalsesteeg 3, 6708 PB Wageningen, The Netherlands; karianne.debruin@gmail.com (K.d.B.); monserrat.budding@wur.nl (M.B.-P.B.)
2   Norwegian Institute for Water Research, Økernveien 94, 0579 Oslo, Norway; Isabel.Seifert@niva.no
3   Netherlands Institute for Ecology, Droevendaalsesteeg 10, 6708 PB Wageningen, The Netherlands; communicatie@nioo.knaw.nl
4   Section of Physical Geography and Hydrology, University of Oslo, 0316 Oslo, Norway; lihong2291@gmail.com
*   Correspondence: gert-jan.wilbers@wur.nl

**Abstract:** Cities are challenged by climate change impacts, such as extreme rainfall events that affect conventional urban water management systems via increased sewage water overflows resulting in water quality deterioration and urban floods causing infrastructure damage. Investments in blue–green infrastructure (BGI) are increasingly considered to address these issues. However, these should be cost-effective. In this study, the effectiveness of five different BGI strategies and one grey strategy are assessed for a peri-urban catchment area in Oslo (Grefsen) using a cost–benefit analysis. The strategies include (i) wadis; (ii) green roofs; (iii) raingardens, rain barrels and wadis; (iv) infiltration crates; (v) water squares, and (vi) a separate sewage system. Besides economic effectiveness, the study also aims to identify the proper protection level by comparing cost–benefit ratios and net benefits for 60-min rainfall events occurring once every 5, 20, and 100 years (M5, M20, and M100), concerning both the current situation and under future climate change (using the Representative Concentration Pathway 8.5). The analyses revealed the highest BC ratios for wadis (12.0–17.3), separate sewage systems (7.7–15.1), and a combination of raingardens, rain barrels, and wadis (1.6–2.3). Strategies dimensioned for less frequent but more intensive rainfall events yielded higher BC ratios. Results for infiltration crates were difficult to interpret and were found to be very sensitive to input parameters. The other strategies implied a negative BC ratio. The study concludes that investments in BGI in Grefsen, Oslo, can be positively judged from a social–economic perspective and provide suitable information for water-related decision makers to decide upon the strategy selection and the appropriate flood protection level.

**Keywords:** climate change; Scandinavia; BGI; economic feasibility

## 1. Introduction

Cities are challenged by climate change impacts, such as extreme precipitation events that challenge the conventional urban water management systems. Stormwater, the precipitation that runs off from impermeable surfaces, increases the volumes of wastewater to be treated and causes overflows of combined sewer systems—allowing untreated wastewater to enter the environment and in more severe cases, potentially damaging urban infrastructure. There is an ongoing transition towards more sustainable urban water management, creating green, climate-adapted, and flood-resilient cities in several cities worldwide [1,2]. Investments in blue–green infrastructure (BGI) measures

are needed to accommodate this transition. BGI measures are strategically planned networks of natural and semi-natural areas with other environmental features designed and managed to deliver a wide range of ecosystem services [3].

Insight into costs and benefits of BGI will help decision makers to assess the effectiveness of different (combinations of) BGI strategies and allocation of scarce financial resources [4,5]. Adaptation pathways provide transition paths in the context of long-term decision making, sequencing measures over time and allowing for adaptive implementation of BGI strategies [6].

Analysis of feasible BGI strategies requires the assessment of climate change impacts, the design and identification of associated costs, and the full range of benefits of BGI measures. Social cost–benefit analysis (SCBA) is often applied to rank and quantify BGI measures under different climate change scenarios, such as different precipitation events [4]. Most SCBAs in the context of sustainable urban water management and BGI options focus on one type of solution such as green roofs [7–9], green walls [10,11] or rainwater harvesting systems [12,13]. Knowledge of associated costs and benefits of BGI measures and the applicability of decision-support tools is increasing, moving towards assessing combinations of measures. Liu et al. [14], for example, conducted a cost–benefit analysis for low-impact development for stormwater reduction in Beijing, China, where the integrated measures showed positive benefit–cost ratios (>1). Johnson and Geisendorf [15] assessed the benefits and economic value of sustainable urban drainage systems at a neighborhood level in Berlin, Germany. In this study, a combination of different BGI measures was considered, and benefits were assessed through ecosystem services. Locatteli et al. [16] presented an economic analysis of green infrastructure applied to two case studies in Spain. They considered a wide range of benefits, ranging from flood damage reduction and water quality improvements to added aesthetic value and considered a combination of different types of GI. However, case studies focusing on BGI in Scandinavian countries are still scarce to date. Furthermore, to our knowledge, there are very few studies that compare cost–benefit ratios of BGIs between different rainfall events and climate change effects. Therefore, there remains a need to gain further insight into the socio-economic and environmental feasibility of the implementation of combinations of measures, especially through case studies that contribute to the knowledge base on the economic aspects of the transition towards sustainable urban water management.

In this study, we present a method for comparing different BGI strategies that are, from an economic perspective, suitable to make the transition towards a blue–green stormwater management system. We assess the costs and benefits of different investment strategies for a case study in a peri-urban sub-catchment in Oslo, Norway. We apply SCBA to assess the net benefits, considering water quality improvements due to prevented sewage water overflows, prevented flood damage benefits, biodiversity increase, increased house prices, prevented sewage water treatment, and avoided tap water use for watering gardens on the benefit side, as well as investment and maintenance costs. Costs and benefits of the transition towards a blue–green stormwater management system are identified and where possible quantified. Further, we identify costs and benefits of BGI strategies for current and future precipitation events taking into consideration expected climate change effects. As such, this economic analysis contributes to the ambitions of the municipality of Oslo to deal with stormwater problems and create a more resilient water system and define appropriate protection levels [17]. Green measures (e.g., green roofs) also increase urban biodiversity and create more green spaces in the city. This aligns with the Oslo municipality objective to increase the area of green spaces in the city [18]. Moreover, the results of this study are useful for other cities world-wide that have to address stormwater management and greening of cities within the scope of climate change.

## 2. Materials and Methods

In this study we assess the benefits and costs of blue–green investments for a peri-urban sub-catchment in Oslo, Norway. This sub-catchment belongs to the larger catchment of the Akerselva River and is located in the northern part of Oslo. It covers most of Grefsen and part of the Grefsen–Kjelsås districts. For simplification, we refer in this paper to the study site as Grefsen. Grefsen is a peri-urban residential area with approximately 6500 inhabitants [19] with a surface area of 1.33 km² (Figure 1). The average elevation of Grefsen is 183 m above sea-level with some slope terrain [20]. The average annual rainfall is 763 mm [21]. The sewage system in Grefsen consists of a mix of combined and separate sewers. Several times a year, especially in the summer months, combined sewer overflows (CSOs) happen during severe rainfall events, discharging surplus water into the Akerselva River via an overflow called AK52. In 2017 for example, a CSO lasted more than five hours. The CSO events have negative impacts on both socio- and environmental conditions. Firstly, the CSOs cause chemical- and biological water pollution, deteriorating downstream water quality in the Akerselva River and the Oslo Fjord, which are used for bathing, thus implying a direct health risk. In addition, the water quality deterioration by CSOs causes mortality of aquatic fauna [22]. Moreover, the current combined system has insufficient capacity to address severe rainfall events, which leads to urban flooding and damage to buildings and other infrastructure.

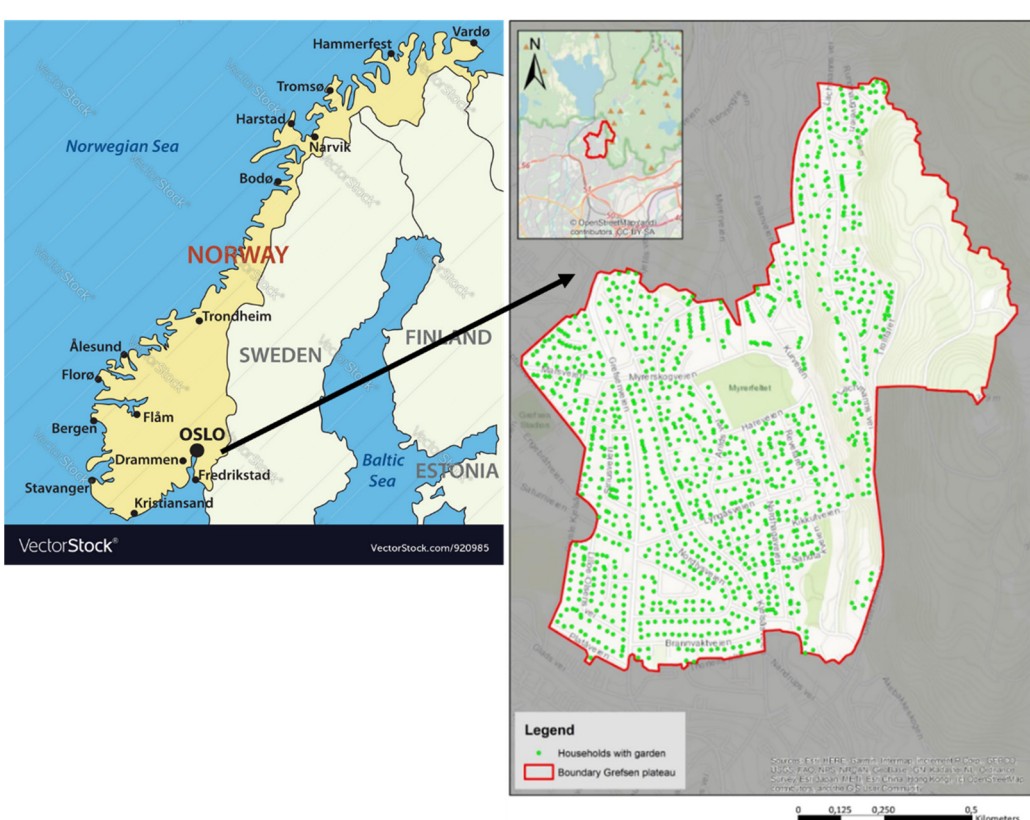

**Figure 1.** Map of the Grefsen study site located in the northwestern part of Oslo, Norway.

### 2.1. BGI Strategies

The city government of Oslo has the ambition to manage stormwater in a way that not only avoids its negative consequences, but also strategically uses BGI options, which provide additional benefits to its inhabitants [23]. The money spent on stormwater management should be used in the most cost-effective way; thus, the city administration is interested in assessing costs and benefits of green and blue infrastructure and combinations of those. Nevertheless, guidelines with respect to maximum occurrence of



CSOs or urban flooding are lacking. The area of Grefsen has already for several years been the focus of the city administration, aiming to reduce the number of CSOs.

In this study, we assessed six BGI strategies to protect Grefsen from 60-min rainfall events occurring once every 5, 20, and 100 years (M5, M20, and M100), as shown in Table 1a. The amount of investments for each strategy were dimensioned to prevent CSOs and urban flooding for the M5, M20, and M100 rainfall events. As such, more investments are required for M100 rainfall events compared to M20 and M5; see Table 1a. Strategies were defined based on the application of one type of BGI or combinations of BGI measures and compared to the current situation consisting of a combined sewer system. Combinations of BGIs were chosen when individual measures lacked insufficient capacity to prevent CSOs and urban flooding. A full description of the selected BGI measures and the associated water storage capacity in Oslo are available in the assessment done by Lekkerkerk [24]. A brief description of BGI measures and their water storage potential within the Oslo context is presented in Table 1b. The Business as Usual (BAU) situation is defined as a situation with no stormwater measures and is used as a reference when deriving and describing benefits as part of the SCBA. For BAU and the BGI strategies, both current rainfall events and future simulated rainfall events based on climate change effects under the RCP 8.5 emission scenario were taken into consideration. However, further effects of temperature, sea level rise, wind, and solar radiation are not considered in this study, although they could have a relevant impact on green infrastructure measures [25]. In total, six BAU and 36 BGI strategies were derived for the different rainfall events under the current climate (referred to as current rainfall events) and anticipated climate change (referred to as future rainfall events).

**Table 1.** (**a**) BGI strategies for different rainfall events for both current and future situations considering climate change. (**b**) BGI description and storage potential within the Oslo context [24].

| (a) | | | | | | |
|---|---|---|---|---|---|---|
| **Strategies** | **Current Rainfall Events (2020)** | | | **Future Rainfall Events (RCP 8.5)** | | |
| | **M5** | **M20** | **M100** | **M5** | **M20** | **M100** |
| Wadis | 479 m² Wadis | 1092 m² Wadis | 1692 m² Wadis | 791 m² Wadis | 1486 m² Wadis | 2835 m² Wadis |
| Green roofs | 9685 m² Green roofs | 22,063 m² Green roofs | 34,185 m² Green roofs | 15,970 m² Green roofs | 22,063 m² Green roofs | 57,275 m² Green roofs |
| Green / blue | 459 Raingarden | 681 Raingarden 601 Rain barrels | 681 Raingarden 681 Rain barrels 343 m² Wadis | 681 Raingarden 125 Rain barrels | 681 Raingarden 681 Rain barrels 343 m² Wadis | 681 Raingarden 681 Rain barrels 1692 m² Wadis |
| Infiltration crates | 151 Infiltration crates of 1 m³ | 344 Infiltration crates of 1 m³ | 533 Infiltration crates of 1 m³ | 249 Infiltration crates of 1 m³ | 468 Infiltration crates of 1 m³ | 893 Infiltration crates of 1 m³ |
| Water square | 1 Water square | 1 Water square | 1 Water square | 1 Water square | 1 Water square | 1 Water square |
| Separate sewer system | Separate sewer system (151 m³) | Separate sewer system (344 m³) | Separate sewer system (533 m³) | Separate sewer system (249 m³) | Separate sewer system (468 m³) | Separate sewer system (893 m³) |

| (b) | | |
|---|---|---|
| **Measures** | **Description** | **Water Storage Potential for Grefsen, Oslo (m³)** |
| Rain garden (per garden) | Residential gardens with a top surface of 2 m², bottom surface of 1 m², a slope of 1:3, and an infiltration box of 1 m³ | 224 |
| Green roofs | Suitable for all roofs with slope <35 degrees. Storage capacity is 15.5 mm due to almost permanent moisture conditions | 2214 |
| Wadis | Properties similar to rain gardens but without a slope, leaving a storage of 0.368 m³/m² | 7490 |
| Rain barrel | Storage capacity of a single rain barrel is 0.2 m³ | 136 |

| | | |
|---|---|---|
| Infiltration crates | Can be implemented under roads, pavements, and fields with maximum required discharge of 7 m³/s | 2769 |
| Water squares | Two locations that can be transformed into a water square (football field and a green zone) | 15,694 |
| Rainwater sewage pipes | Can be implemented under roads, pavements, and fields with maximum required discharge of 7 m³/s | 2769 |

### 2.2. Precipitation and CSO Volumes

The current rainfall events were derived from intensity–duration–frequency (IDF) curves for 2020, representing the cumulative precipitation of a hypothetical rainfall event with a certain frequency. The IDF curves for this study are based on precipitation data from the Blindern measurement station (approximately 5 km from Grefsen) using data from 1968–2005 [26,27]. Precipitation events with a duration between 0–60 min with a 5-min time interval were used. Future rainfall events were calculated by applying climate factors (CFs) to current rainfall events using the Representative Concentration Pathway (RCP) scenario 8.5, which is the highest emission concentration climate scenario [28] and is regarded as a worst-case climate scenario in this study. The climate factor (CF) for RCP 8.5 in the Norwegian context was computed by Dyrrdal & Forland [29], who derived CFs for precipitation durations of 1, 3, 6, and 12 h with a return interval between five and 200 years. The median computed climate factor for the year 2050 with a precipitation duration of 1 h was selected and is 1.18, 1.19, and 1.21 for 5-, 20-, and 100-year return intervals, respectively. The climate factor for RCP 8.5 for the year 2100 was selected to provide a bandwidth between no climate effects (current rainfall events) and a worst-case event (taking into consideration expected rainfall under RCP 8.5 in the year 2100).

CSOs at Grefsen occur when the capacity of the sewage system exceeds 600 l/s at the overflow AK52 [24]. The CSO volumes of the AK52 sewage overflow were derived through modelling 60-min rainfall events with different return periods of 5, 20, and 100 years in the study area applying the Stormwater Management Model (SWMM) [24]. The model simulated the discharge for every precipitation event, and all discharges greater than 600 l/s were regarded as the water surplus leaving the system via the overflow AK52. Table 2 presents the precipitation and CSO volumes for both current and future rainfall events for M5, M20, and M100.

**Table 2.** Precipitation (mm) and sewage overflow volumes (m³) for Grefsen at the overflow AK52 during multiple rainfall events under current (2020) and future (RCP 8.5) scenarios.

| Rainfall Event | Current Rainfall Events (2020) | | Future Rainfall Events (RCP 8.5) | |
|---|---|---|---|---|
| | 60-min Precipitation (mm) | CSO Volume (m³) | 60-min Precipitation (mm) | CSO Volume (m³) |
| M5 | 25.1 | 151 | 29.6 | 249 |
| M20 | 34.7 | 344 | 41.3 | 468 |
| M100 | 45.3 | 533 | 54.8 | 893 |

### 2.3. Costs

The Capital (CAPEX) and Operation and Maintenance (OPEX) costs as well as the economic lifetime of the green, blue, and grey measures are based on a literature research and expert judgment carried out in Lekkerkerk [24]. The input data were verified with stakeholders of Oslo municipality. An overview of unit prices in Norwegian Krones and EURO (converted using a six-month average exchange rate from 23 October 2020–22 April 2021 (Exchange-rates.org) (accessed on 22 April 2021)) and the economic lifetime for the different measures are presented in Table 3.

**Table 3.** CAPEX, OPEX, and economic lifetime of the green, blue, and grey measures. The water square entails a fixed transition of a grass field into a water square. This dimension was applied in all scenarios and rainfall events.

| Measures | CAPEX NOK (€) | OPEX per Year NOK (€) | Economic Lifetime (Years) |
|---|---|---|---|
| Wadis (per m2) | 505 (49) | 0.7 (0.06) | 50 |
| Green roofs (per m2) | 814 (78) | 12 (1.2) | 50 |
| Rain gardens (per garden) | 16,446 (1581) | 250 (24) | 50 |
| Rain barrels (per barrel) | 915 (88) | 250 (24) | 20 |
| Infiltration crates (per m3 storage capacity) | 14,301 (1375) | 1.4 (0.14) | 50 |
| Water square (11,525 m2 fixed) | $28.4 \times 10^6$ ($2.73 \times 10^6$) | 48,000 (4615) | 70 |
| Rainwater sewage pipes (per m3 storage capacity) | 1235 (119) | 0.2 (0.02) | 70 |

Annual maintenance efforts are needed to reach economic lifetime of measures. Nevertheless, it is assumed that after lifetime exceedance, the measures will no longer be suitable, and re-investments (in the CAPEX) are required. According to the European recommendations for evaluation of investments [30], the residual values of the measures should be taken into consideration in the CBA. This value reflects the remaining use of the measure after the calculation period (30 years in this study) and is included as a negative cost under CAPEX in the last evaluation year.

Via expert judgment and consultation with the Oslo city administration, the phasing of investments was defined. For wadis, green roofs, raingardens, and rain barrels, a 5-year investment period was chosen as these will be implemented mainly by private landowners through stimulation programs over multiple years. Infiltration crates and a water square imply an investment period of one year as they are regarded as a one-time investment in this study. The installation of rainwater sewage pipes is expected to have a 10-year investment period as these will be installed during regular maintenance work on the mixed sewage system to minimize the breaking up of streets.

*2.4. Benefits*

Investments related to BGI measures in Grefsen aim to prevent CSOs and urban floods, and thus, avoidance of CSOs and urban flooding are regarded as direct benefits of these investments. In addition, the measures provide multiple co-benefits that would not be achieved by applying grey measures. Co-benefits of green infrastructure are related to enhanced environmental soundness, improved public health, and other improvements (e.g., fresh water savings) to the built environment [31,32]. In this study, both direct and co-benefits of the measures are considered and compared with the BAU situation as this is typically done in CBAs [16].

2.4.1. Direct Benefits

Prevented sewage water overflow (CSO) is the main objective of implementing BGI measures. Reduction or prevention of CSO contributes to improved water quality that positively impacts the Akerselva River biodiversity due to reduced pollution. Further, health benefits are expected as downstream locations of the Akselva River are used for bathing. However, quantification and monetarization of these benefits was not executed in this study. The overflow AK52 at Grefsen is only one of multiple overflows in Oslo that discharge in the Akerselva River, and thus it was not possible to disentangle changes in downstream biodiversity effects and bathing water quality from other overflows. Nevertheless, preventing CSOs from AK52 will have a positive contribution to both biodiversity effects and bathing water quality. Therefore, the increased biodiversity and effects on

bathing water quality of the Akerselva River are included qualitatively (with +/−) using expert judgment.

Prevented flood damage is derived in two steps: (1) development of flood maps with SIMulated Water Erosion (SIMWE) for Grefsen [33]. The SIMWE model uses terrain, infiltration, and surface roughness as main inputs, and flood depths were derived for M5, M20, and M100 rainfall events for the current and future situations and (2) damages were then derived for buildings (houses, garages, carports) using the damage functions applied in urban areas of Oslo [34]. The damage functions use a threshold of 3 cm, i.e., damages to buildings only occur when flood levels exceed 3 cm above ground level. A citizen panel in Grefsen indicated flooding damages to occur to garages, carports, and basements (houses) in periods of excess rainfall. As such, damages to these infrastructural types have been included in the damage calculations. Damage to other infrastructure (roads, vehicles) and business interruptions was not considered. Damages were derived for the BAU situation for each (current and future predicted) rainfall event. As measures are dimensioned to prevent urban floods for a specific protection level (M5, M20 or M100), it is expected that no urban floods will occur up to the 100-year 60-min rainfall event (in the case of M100) after full implementation of the BGI strategies. As such, the derived damage values at BAU for the different rainfall events are regarded as prevented damage values in the SCBA.

### 2.4.2. Co-Benefits

The realization of green infrastructure in general (e.g., green roofs, raingardens, and wadis) increase the green character of the neighborhood that entails an increase in the aesthetical value and has therefore been included as a co-benefit. Further, co-benefits include increased house prices due to installation of green roofs [35,36]. Precipitation from roofs, streets, and other infrastructure, which is currently collected in the combined sewage system and transported to the waste water treatment plant (WWTP) comes with a treatment cost of 13 NOK/m$^3$ in the BAU [37]. The reduction of the amount of water flowing to the WWTP due to BGI measures- urther mentioned as prevented sewage water treatment)- is therefore regarded as a co-benefit. Furthermore, the application of rainwater barrels leads to decreased tap water use for watering gardens and is included as a co-benefit.

A full description of the calculations and assumptions made to derive the co-benefits is presented in the Supplementary Materials information [38–43].

### 2.5. Cost–Benefit Calculations

The outcomes are presented by deriving the net benefits of the strategies (Million NOK) and the benefit/cost ratio per strategy as presented in Table 1a. Costs and benefits were derived from the net present value (NPV), applying a discount rate of 4% following guidelines of the Norwegian government [44]. A reference period of 30 years was selected, which is in line with the European Commission CBA guideline for water-related projects [45].

For each year, the total benefits and costs were derived and cumulatively summed over time. The BC ratio was then defined after each project year (n1–30). Based on this assessment, the return on investment period of BGI strategies was visible through graphs for the year that the BC ratio exceeded 1.

## 3. Results

### 3.1. Costs

Tables 4 and 5 present the CAPEX and OPEX for the six BGI strategies for current and future rainfall events, respectively, calculated as the net present value of the stream of costs over a 30-year time horizon. The tables show that the green roof strategy implies higher costs compared to most other strategies due to required large investments during installation, as well as associated maintenance activities to guarantee optimal functioning of the roofs. In contrast, the separate sewage system strategy is the cheapest alternative although the implementation time is long as the separate sewage systems are only installed in Oslo (and most other cities) during regular infrastructural maintenance activities in the city. The costs for the water square are equal for all current and future rainfall events as this alternative implies a fixed transition from an open grass field to a water square. As expected, the CAPEX/OPEX of BGI strategies was greater for future rainfall events (RCP 8.5) compared to current events, as measures have to deal with more water compared to the dimensions of current rainfall events.

**Table 4.** Net Present Value of Costs, Benefits, and Co-Benefits of the different investment strategies under current rainfall events in 2020 (million NOK).

| Strategy | Precipitation Event | Costs | | | Direct Benefits | | Co-Benefits | | | | Total Benefits |
| | | CAPEX | OPEX | Total Costs | River Biodiversity and Water Quality Effects ** | Prevented Flood Damage | Increased Aesthetical Value (Greener Oslo) | Increased House Prices | Prevented Sewage Water Treatment | Fresh Water Savings | |
|---|---|---|---|---|---|---|---|---|---|---|---|
| Wadis | M5 | 0.17 | 0.01 | **0.18** | + | 0.52 | 0.24 | | 1.52 | | **2.28** |
| | M20 | 0.40 | 0.01 | **0.41** | ++ | 1.68 | 0.54 | | 3.47 | | **5.69** |
| | M100 | 0.61 | 0.02 | **0.63** | +++ | 4.68 | 0.83 | | 5.38 | | **10.89** |
| Green roofs | M5 | 5.67 | 1.62 | **7.29** | + | 0.52 | 0.80 | 3.46 | 0.55 | | **5.33** |
| | M20 | 12.93 | 3.70 | **16.63** | ++ | 1.68 | 1.82 | 7.89 | 1.26 | | **12.65** |
| | M100 | 20.05 | 5.73 | **25.78** | +++ | 4.68 | 2.82 | 12.22 | 1.96 | | **21.68** |
| Green/blue | M5 | 5.43 | 1.61 | **7.04** | + | 0.52 | 7.88 | | 3.94 | 0.00 | **12.34** |
| | M20 | 8.64 | 4.48 | **13.12** | ++ | 1.68 | 11.69 | | 7.57 | 0.55 | **21.49** |
| | M100 | 8.92 | 4.77 | **13.69** | +++ | 4.68 | 11.96 | | 9.55 | 0.62 | **26.81** |
| Infiltration crates | M5 | 1.73 | 0.00 | **1.73** | ++ | 0.59 | | | 0.62 | | **1.20** |
| | M20 | 3.94 | 0.01 | **3.95** | +++ | 1.89 | | | 1.40 | | **3.29** |
| | M100 | 6.10 | 0.01 | **6.11** | ++++ | 5.26 | | | 2.17 | | **7.43** |
| Water square | M5 | 21.45 | 0.75 | **22.20** | ++ | 0.59 | | | | | **0.59** |
| | M20 | 21.45 | 0.75 | **22.20** | +++ | 1.89 | | | | | **1.89** |
| | M100 | 21.45 | 0.75 | **22.20** | ++++ | 5.26 | | | | | **5.26** |
| Separate sewer system | M5 | 0.11 | 0.00 * | **0.11** | 0/+ | 0.45 | | | 0.52 | | **0.97** |
| | M20 | 0.25 | 0.00 * | **0.25** | + | 1.45 | | | 1.19 | | **2.64** |
| | M100 | 0.39 | 0.00 * | **0.39** | ++ | 4.05 | | | 1.84 | | **5.89** |

* The table shows rounded values, although the actual value >0.00 million NOK. ** Effects of CSO prevention on Akerselva River biodiversity and bathing water quality were determined qualitatively. Note: total costs and benefits are presented in bold.

**Table 5.** Net Present Value of Costs, Benefits, and Co-Benefits of the different investment strategies under future rainfall events with anticipated climate change (RCP 8.5) (million NOK).

| Strategy | Precipitation Event | Costs | | | Direct Benefits | | Co-Benefits | | | | |
| | | CAPEX | OPEX | Total Costs | Effects on CSO's * | Prevented Flood Damage | Increased Aesthetical Value (Greener Oslo) | Increased House Prices | Prevented Sewage Water Treatment | Fresh Water Savings | Total Benefits |
|---|---|---|---|---|---|---|---|---|---|---|---|
| Wadis | M5 | 0.29 | 0.01 | **0.30** | + | 0.62 | 0.39 | | 2.51 | | **3.52** |
| | M20 | 0.54 | 0.01 | **0.55** | ++ | 2.11 | 0.73 | | 4.73 | | **7.57** |
| | M100 | 1.03 | 0.03 | **1.06** | +++ | 6.45 | 1.39 | | 9.02 | | **16.86** |
| Green roofs | M5 | 9.36 | 2.68 | **12.04** | + | 0.62 | 1.32 | 5.71 | 0.91 | | **8.56** |
| | M20 | 17.59 | 5.03 | **22.62** | ++ | 2.11 | 2.47 | 10.73 | 1.72 | | **17.03** |
| | M100 | 33.56 | 9.59 | **43.15** | +++ | 6.45 | 4.72 | 20.48 | 3.28 | | **34.93** |
| Green/blue | M5 | 8.18 | 2.82 | **11.00** | + | 0.62 | 11.69 | | 6.21 | 0.11 | **18.63** |
| | M20 | 8.84 | 4.77 | **13.61** | ++ | 2.11 | 11.86 | | 8.89 | 0.62 | **23.48** |
| | M100 | 9.33 | 4.78 | **14.11** | +++ | 6.45 | 12.53 | | 13.18 | 0.62 | **32.78** |
| Infiltration crates | M5 | 2.85 | 0.01 | **2.86** | ++ | 0.70 | | | 1.02 | | **1.71** |
| | M20 | 5.35 | 0.01 | **5.36** | +++ | 2.37 | | | 1.91 | | **4.28** |
| | M100 | 10.22 | 0.02 | **10.24** | ++++ | 7.24 | | | 3.64 | | **10.88** |
| Water square | M5 | 21.45 | 0.75 | **22.20** | ++ | 0.70 | | | | | **0.70** |
| | M20 | 21.45 | 0.75 | **22.20** | +++ | 2.37 | | | | | **2.37** |
| | M100 | 21.45 | 0.75 | **22.20** | ++++ | 7.24 | | | | | **7.24** |
| Seperate sewer system | M5 | 0.18 | 0.00 * | **0.18** | 0/+ | 0.54 | | | 0.86 | | **1.40** |
| | M20 | 0.34 | 0.00 * | **0.34** | + | 1.82 | | | 1.61 | | **3.43** |
| | M100 | 0.65 | 0.00 * | **0.65** | ++ | 5.57 | | | 3.08 | | **8.65** |

* The table shows rounded values, although the actual value >0.00 million NOK. Note: total costs and benefits are presented in bold.

### 3.2. Direct and Co-Benefits

Tables 4 and 5 present the direct- and co-benefits in NPV (million NOK) for the different BGI strategies under the current and future rainfall events, calculated as the net present value of the stream of direct and co-benefits over a 30-year time horizon. Note that the effects on the Akerselva River biodiversity and bathing water quality improvements due to CSO prevention were determined qualita20:tively with +/− scores. A deviation in ranking is based, besides the dimensioning of measures (M5, M20 or M100), on the implementation time that differs between strategies.

The avoided flood damages are presented in Figure 2 for both current and future rainfall events, based on flood damage functions for Oslo, Norway [34], and the chance of occurrence. The curves were developed through linear interpolation between M5, M20, and M100. The prevented flood damage for the different BGI strategies was derived from the area below the curves that corresponds to the M5, M20, and M100 rainfall events. For M100 rainfall events, the total surface area as presented in Figure 2 was regarded. In the SCBA model, the prevented flood damage value per rainfall event was calculated per year to account for the implementation duration of the strategies. This explains the varying prevented flood damage values of the total project period between strategies for the same rainfall events.

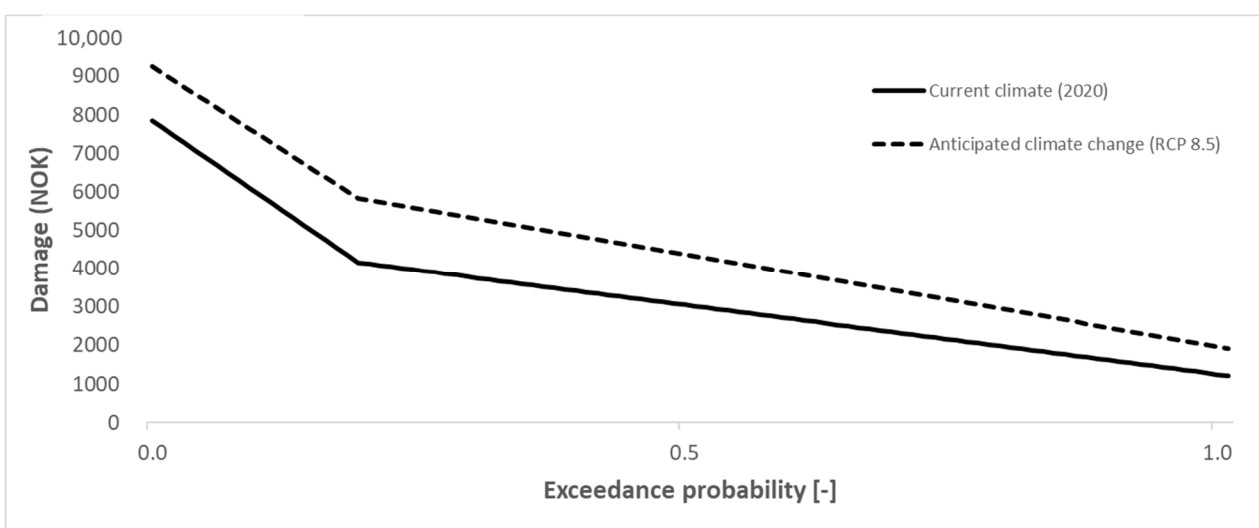

**Figure 2.** Flood damage curves for different rainfall exceedance probabilities for the current situation (no climate change) and future situation considering climate change effects (RCP 8.5).

The green aesthetical value was highest for the green/blue strategies as these entail most green investments compared to other BGI strategies. Increased house prices were obtained only for strategies including green roofs while fresh water saving was only included in the green/blue strategy due to inclusion of rainwater barrels. Prevented sewage water treatment was highest for the green/blue strategy as it contains the most green and blue spaces among all strategies, allowing for the maximum rainwater capture area and thus prevention of run-off to the combined sewage system.

### 3.3. Net Benefits and Return on Investment

The discounted benefit/cost (BC) ratio and the net benefits were derived per strategy under different rainfall events between the current (2020) and future situation considering climate change and are presented in Figure 3, using a 30-year time horizon. The BC ratio shows to what extent the net present value (NPV) of benefits outweighs the NPV of the costs while the net benefit indicates the monetarized total benefits of the BGI strategies. Considering the BC ratio, wadis, separate sewer systems, and green/blue strategies ranked the highest. From a net benefit perspective, green/blue strategies ranked the highest followed by wadis and separate sewer systems. The other strategies imply a BC ratio lower than 1 except for infiltration crates dimensioned for a M100 rainfall event. This implies that decision makers aiming for benefit maximalization from investments may prefer wadis while decision makers interested in maximizing the net monetarized benefit may prefer the green/blue strategy. Moreover, for wadis, green/blue strategies, and separate sewer systems, both the BC ratio and net benefits were greater when dimensioned for higher rainfall intensities. This was, however, less visible for the other strategies.

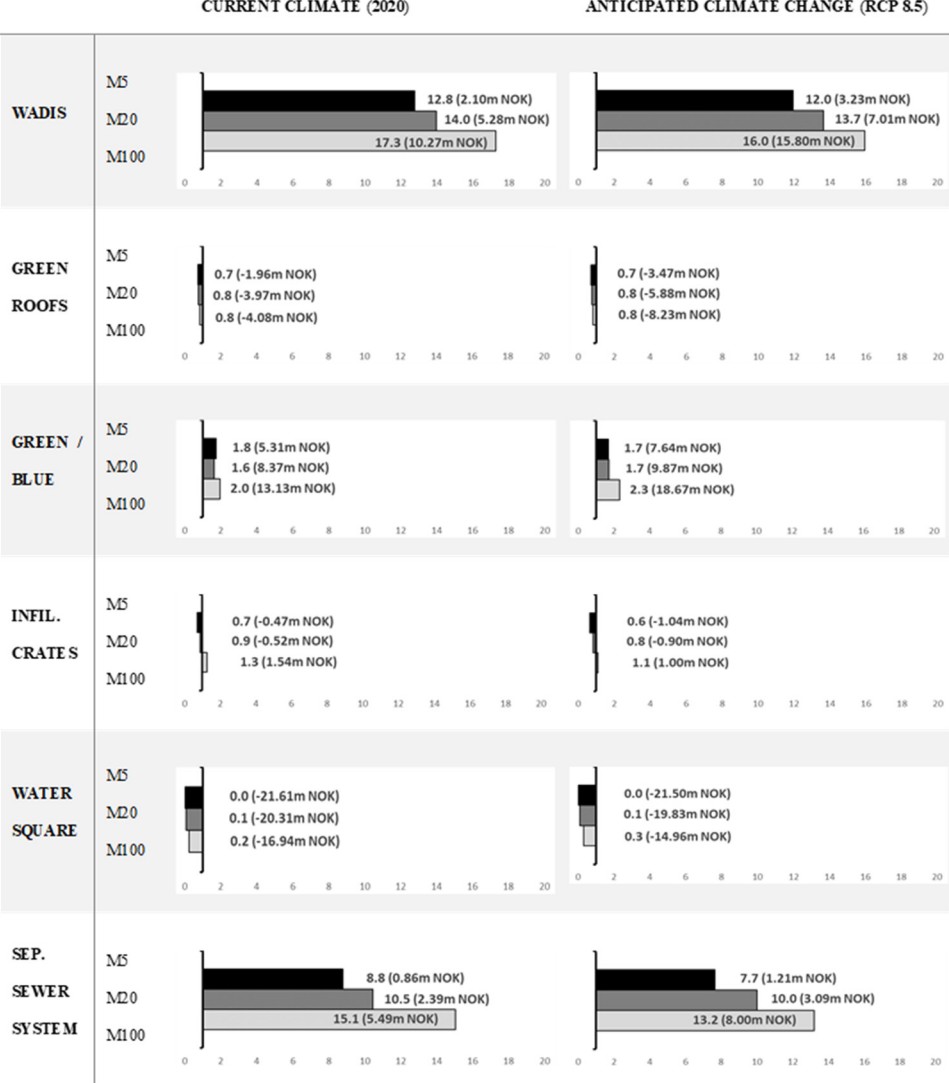

**Figure 3.** Benefit/cost ratio (on the horizontal axis) and net benefit for the various strategies (net benefit value presented in parentheses).

The development of BC ratios over time for all strategies under M5 (future situation) and M100 (future situation) was plotted to define the return on investment (ROI) period; see Figure 4. The presentation shows the range of ROI periods for strategies under different rainfall events. Future rainfall events presented the worst-case scenario. Return on investment was achieved when the BC ratio reached values >1. For wadis, this point was reached between 4–5 years while it required between 5–7 years for the separate sewer system strategy and 10–14 years for the green/blue strategy. The infiltration crate strategy had a return on investment period of 18 years when designed for M100 rainfall events, although the break-even point exceeded 30 years in the case dimensioned for M5. The water square and green roofs strategies did not generate BC ratios higher than 1 over the 30-year period for both M5 and M100 rainfall events. This information is useful for decision makers to identify preferred strategies and rainfall event protection levels when return on investment is seen as an important parameter in decision making.

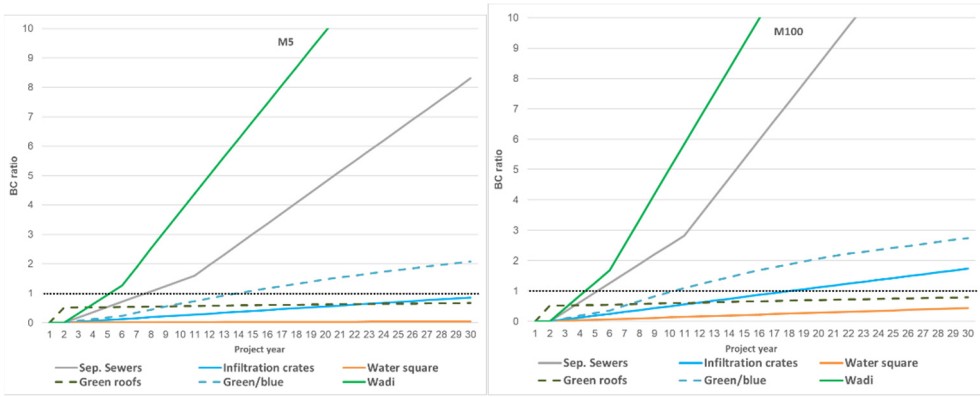

**Figure 4.** Development of BC ratios over 30 years (without considering the residual values of investments after 30 years) for the six strategies under M5 and M100 rainfall events for the future situation. Lines crossing the dashed line shows the return on investment period.

### 3.4. Sensitivity Analyses

Sensitivity analyses were performed to address uncertainty in input parameters. In this study, six sensitivity assessments were conducted for each strategy for the M100 future situation with the climate factor to present sensitivity of the worst-case rainfall event scenario. Effects on the CB ratio are presented. The sensitivity parameters included (i) costs (CAPEX + OPEX); (ii) the discount rate; (iii) prevented flood damage benefit; (iv) aesthetical benefits of increased green areas; (v) prevented sewage water treatment benefits; and (vi) increased house prices from green roofs. Sensitivity was assessed by applying two factors for these parameters (factors 0.5 and 2) that were compared with the reference situation (factor 1) and are graphically presented in Figure 5. Effects on the CB ratio for the entire project period (30 years) were assessed.

The analyses revealed the largest variations in the BC ratios for the wadis and separate sewer system strategies. As these strategies have relatively lower CAPEX and OPEX compared to the other strategies, these are more susceptible to changes in costs, discount rates, and benefits. Nevertheless, sensitivity assessment revealed no shift in the BC ratios <1.0 for these strategies when costs, discount rates, and benefits were either reduced or increased by a factor of 2, which indicates their solid cost-effectiveness. This shows that data uncertainty mostly did not significantly impact the BC ratios, as this would imply a change from a net positive result to a net negative result or vice versa. However, the water square strategy was an exception in this context, as the BC ratio in the reference situation (thus without sensitivity analyses) of 1.1 decreased to <1.0 when CAPEX/OPEX and the discount rate were doubled (factor 2) and when prevented flood damage and prevented sewage water treatment benefits were reduced by a factor of 2. Furthermore, for green roofs, the BC ratio increased to 1.3 from 0.8 (in the reference situation) when the increased house price benefits doubled.

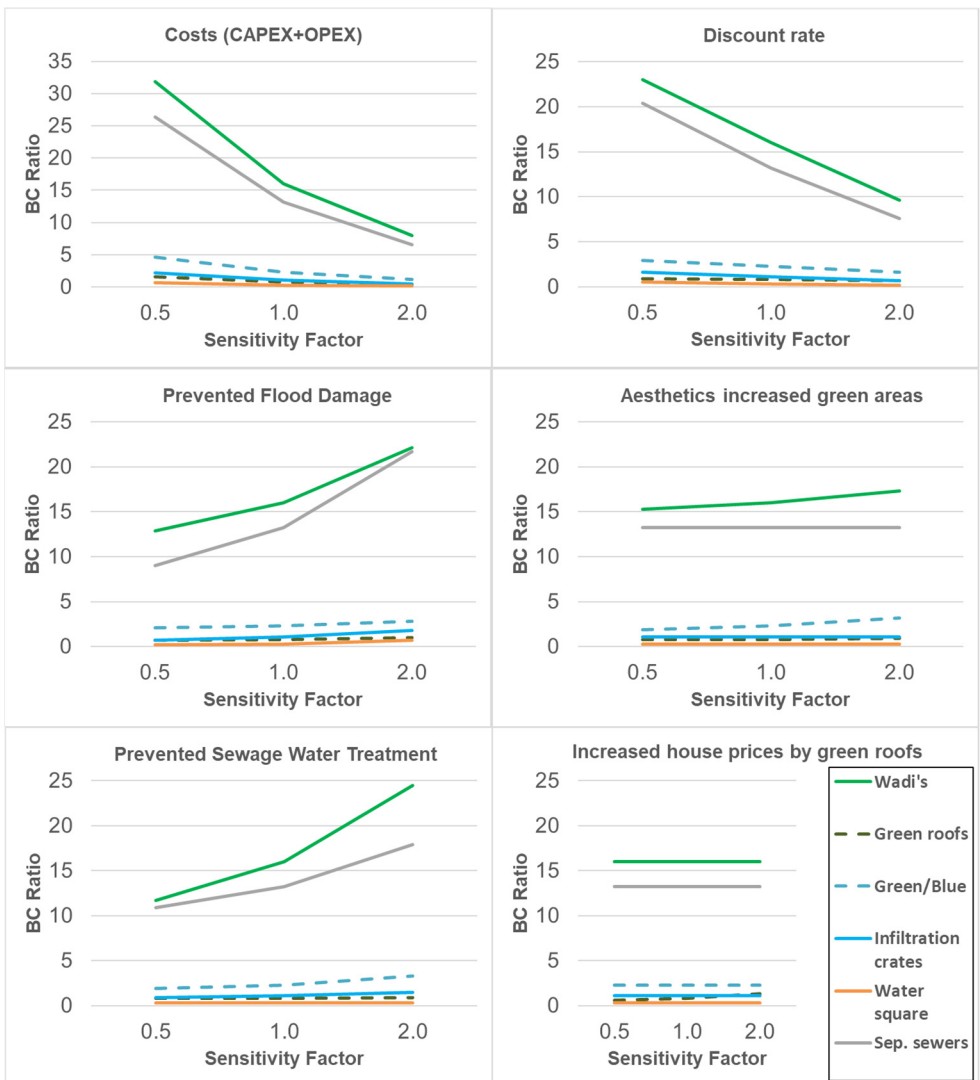

**Figure 5.** Sensitivity analyses of BC ratios for the six strategies (M100; future situation) for costs and discount rates, as well as prevented flood damage-, aesthetic-, prevented sewage water-, and increased house price benefits. Factor 1 represents the reference situation (without sensitivity assessment). Note the different y-axis scale on the left top graph for costs (maximum of 35 instead of 25).

## 4. Discussion

This study presented an analysis of the costs and benefits of investing in BGI strategies to reduce sewer overflows and improve water quality. Costs included investment costs and maintenance costs, and benefits included both direct benefits and co-benefits. The results of the SCBA showed that wadis and separate sewer systems resulted in the highest BC ratio and fastest return on investment, although from a net benefit perspective, the green/blue strategy ranked highest. In addition, the study revealed higher BC ratios and greater net benefits for strategies dimensioned for M100 rainfall events compared to M20 and M5 events. Furthermore, from a cost–benefit perspective, strategies dimensioned for rainfall events considering climate change effects had higher BC ratios and greater net benefits compared to current rainfall events. This suggests that BGI strategies should be dimensioned for less frequent but high rainfall events (e.g., M100) from a socio-economic and environmental perspective. However, it should be noted that other factors may also be relevant in selecting appropriate strategies besides the cost–benefit analysis. For example, governments may prefer strategies to be installed under private–public partnership (PPP) to reduce investment and maintenance costs. In such a case, strategies including measures such as rain barrels and green roofs may be preferred. Moreover, local

governments need to focus increasingly on expanding green spaces for improving sustainability and livability in cities [46] and may thus be a main criterion for the selection of strategies as well. Furthermore, criteria such as required implementation duration of strategies, participation or preferences of citizens, and coupling possibilities with existing infrastructural maintenance works may be relevant in the selection of strategies. If such decision criteria are to be applied, it is recommended to additionally use a multi-criteria analysis (MCA) [47]. This CBA can then be a part of the MCA.

As pointed out by Locatelli et al. [16], CBAs are sensitive to the availability of data on the costs and benefits, uncertainties, and model assumptions. In our case, we found that BC ratios were relatively sensitive when input data were increased and/or decreased by a factor 2 (up to even 100%). However, sensitivity analyses also revealed that these uncertainties mostly did not lead to a shift from a net positive to a negative net result for most strategies. This was not the case for the water square strategy that indicated that the results of this study should be treated with care and that particular benefits should be assessed in more detail to determine the cost-effectiveness of a water square in Grefsen. Furthermore, green roofs shifted from a net negative result in the reference situation towards a positive result when the increased house price benefit was doubled. Therefore, effects of green roof installation on house price increases in Grefsen specifically should be validated (i.e., via real estate transactions of houses with and without green roofs in Oslo or other Norwegian cities) to better quantify this benefit. However, this level of detailed information was not available.

In our analysis, the prevented flood damages due to the implementation of urban blue–green infrastructure were based on theoretical damage models. Theoretical flood damage models are based on threshold levels for floods in houses, commercial buildings, and industries that may not be valid for all specific locations, which could result in severe over- or under-estimation of flood damage. Validation of results in the field (i.e., insurance data) would therefore enrich the accuracy on deriving urban flood damages. Sensitivity analysis of the discount rate showed that the BC ratio may fluctuate from higher than 30 to less than 10 for discount rates between 2% and 8%. However, it should be noted that Norway has set guidelines regarding the selection of the discount rate for CBA [44]. Grefsen is an area, that in the BAU situation, is already characterized as a relative green suburb of Oslo, which could affect the willingness to pay for additional green spaces. However, sensitivity analysis revealed a limited impact of this benefit on the BC ratios of all strategies. Sensitivity analysis related to CAPEX + OPEX and prevented sewage water treatment benefit did not reveal changes from a net positive to a negative result.

The water quality and biodiversity benefit was determined qualitatively, as actual effects of CSO prevention on downstream water quality is difficult to assess. Firstly, other CSO overflow remains in Oslo, with negative results on the downstream water quality. Secondly, CSOs temporarily pollute surface waters (only during peak rainfall events), and detailed water quality data on this level are lacking. Although the effects of removing CSOs from Grefsen on the Akerselva water quality and biodiversity are difficult to assess, there is certainly a positive effect on downstream water quality during extreme rainfall events.

Green roofs were found not to be a cost-efficient strategy. Montalto et al. [48] found that green roofs had lower cost-effectiveness compared to other measures to reduce CSOs in Brooklyn, New York, USA. Nevertheless, our study is in contradiction with a study by Blackhurst et al. [49], who found that green roofs are cost effective when installed on multifamily houses and commercial buildings when all social benefits are included. Moreover, a study in Helsinki concluded that when adding up private and public benefits, the benefits would surpass costs and make green roofs good investments for society [9]. It should be noted that both studies stated that private benefits did not outweigh the costs and that green roofs only became cost effective when multiple social benefits (such as increased lifespan of the roof, energy savings, better air quality, sound insulation, aesthetics, health benefits, stormwater prevention, improved biodiversity, etc.) were included.

However, many of these benefits were not applied in this study. Air quality, for example, is mainly an issue in winter due to emissions resulting from heating houses. During this time, roofs are mostly covered by snow, and thus green roofs have no effect on capturing air pollutants. This shows the relevance of local spatial- and climatic-specific circumstances of a study site.

On the inclusion of climate scenarios, we considered the current situation (based on historic data) and a situation with climate change impacts (RCP 8.5). Although RCP 8.5-associated precipitation events are expected to occur in the future (2050), we assume in the CBA model that these rainfall events occur to date. We chose to do so in order to present a worst-case scenario and to present a range of possible climate effects on extreme rainfall.

Overall, investments in BGI in Grefsen are economically feasible although not for all selected strategies. The lower cost strategies such as wadis had the highest BC ratios compared to other strategies. This finding is in line with that reported by Johnson and Geisendorf [15], who also found that BGI was particularly economically feasible for cheaper options. Moreover, local specific circumstances play an important in the economic feasibility of BGI. A study in Barcelona and Bandalona, for example, revealed that similar BGI strategies in Barcelona were more cost-effective compared to Bandalona due to local conditions such as the severity of urban floods [16]. Thus, at locations where BGI is considered an option to improve urban water management, detailed assessments of potential benefits should be performed, as this determines which type of strategies are feasible.

## 5. Conclusions

This study showed that wadis, a combination of green/blue measures (raingardens, rain barrels, and wadis) and a separate sewer system are cost-effective strategies to address stormwater management in Grefsen, Oslo. Both wadis and green/blue strategies have a larger number of benefits and can be installed within a shorter timeframe compared to the separate sewer system strategy. From a benefit–cost perspective, the analysis revealed higher BC ratios and greater net benefits when strategies were dimensioned for less frequent but more intensive rainfall events; the M100 dimensioned strategy resulted in higher BC ratios and greater net benefits compared to M20 and M5. The approach and results of our analysis are relevant for decision makers responsible for stormwater management to decide on appropriate urban flood protection levels (M5, M20 or M100) based on cost/benefit analysis. The approach also provides insights into the effectiveness of the different strategies, although decision makers have to decide for themselves what they find most important (e.g., net benefits, benefit/cost ratio, return-on-investment period, the variety of (co)benefits). As such, the analysis does not answer which alternative project strategy should be selected but gives the information and tools for a policy dialogue to make the best choice based on the stormwater management objectives of a specific region/city. Moreover, MCA may additionally be needed when other parameters have to be included such as weighting of financial instruments and preference of citizens, among other criteria. The Oslo case study approach is also relevant for other cities worldwide affected by stormwater management issues, although the costs and benefits should always be taken into consideration for specific local conditions.

**Supplementary Materials:** The following are available online at www.mdpi.com/article/10.3390/su14031934/s1, Supplementary data: Defining co-benefits.

**Author Contributions:** Conceptualization, G.-J.W., K.d.B. and I.S.-D.; methodology, G.-J.W. and K.d.B.; software, H.L. and W.L.; validation, G.-J.W., K.d.B., M.B.-P.B. and I.S.-D.; formal analysis, G.-J.W. and M.B.-P.B.; investigation, G.-J.W. and K.d.B.; resources, Wiebe Lekkerkerk, I.S.-D.; data curation, W.L. and H.L.; writing—original draft preparation, G.-J.W. and K.d.B.; writing—review and editing, I.S.-D., M.B.-P.B.; visualization, H.L. and W.L.; supervision, I.S.-D.; project administration, I.S.-D.; funding acquisition, I.S.-D. All authors have read and agreed to the published version of the manuscript.

**Funding:** This research was funded by The Research Council of Norway, grant number 270742 and the APC was funded by Wageningen Environmental Research.

**Institutional Review Board Statement:** Not applicable.

**Informed Consent Statement:** Not applicable.

**Acknowledgments:** The paper is based on research undertaken for the New Water Ways (270742) project funded by the Research Council of Norway. The authors thank Espen Lund (NIVA) for overlaying the flood maps with the buildings and further GIS help.

**Conflicts of Interest:** The authors declare no conflict of interest.

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
