# Peer review of "Investing in Urban Blue–Green Infrastructure—Assessing the Costs and Benefits of Stormwater Management in a Peri-Urban Catchment in Oslo, Norway"

_sustainability, doi:10.3390/su14031934_

Round 1

Reviewer 1 Report

In my opinion, the article needs to improve the following aspects:

Rewrite the keywords to remove the compound ones and reduce them. The summary is too broad and does not express a main conclusion with the title, Where is the quantified cost and benefit of the use of stormwater?. It also needs a discussion of results with similar works, contributed to the bibliography, before the conclusions.

BROAD AND SPECIFIC COMMENTS:

The work seems to me very original and very much in line with the edition of the magazine.

The work must rewrite the keywords to remove the compound ones and reduce them. The summary is too long, needs to be shortened, and does not express a main conclusion with the title, Where is the quantified cost and benefit of stormwater use in the summary?.

The discussion is well done Based on the data obtained, what I can't find is the data source, they should put where they get it from.

And finally add a section or the same of the conclusions that indicates some type of limitation in the study carried out, as well as leaving an open door to future research.

Reviewer 2 Report

sustainability-1576800

The authors should be commended for their work that is a good contribution toward disentangling options that will help deal with current and future urban drainage issues that will almost certainly be exacerbated by the impacts of climate change.  The study is very nicely conceptualized overall, and the analysis is well-executed and explained. We do need an understanding of how portfolios of BGI could be applied in specific geographic locations and what their benefits and costs are likely to be.

There are a couple of notable areas in which this manuscript requires revision, however.

  • The term scenario is inappropriately used to describe strategies for improving water infrastructure performance or to alleviate stress on the urban drainage system. In almost all climate change adaptation literature I have seen, the term scenario is used to describe future conditions (i.e., general climatological conditions or specific aspects of the climate, such as precipitation duration and intensity). Strategies would then be ways of dealing with or addressing the future scenarios.  It is therefore strongly suggested that the word scenario in this manuscript be replaced with the word ‘strategy’ or a similar term, and that if the word scenario is used it should refer to future climate conditions, which would bring this manuscript more in line with what is found in the rest of the literature.

  • Even though selected BGI measures are fully described in the work by Lekkerkerk (2020), there needs to be at least a short table in this manuscript describing these measures and their potential for storage or retention in the Oslo context. Some (e.g., rain barrels) are much more universally deployed and self-explanatory than others (e.g., wadis) which may have more region-specific popularity and uses.

  • There needs to be a clear definition of what BGI are. The definition on lines 47-8 is inadequate.

  • On line 91, there is mention of “bringing nature back to the city”-this is exceedingly vague. What does this mean? Is Oslo entirely devoid of nature (doubtful)?  Some basis of comparison is needed…or at least some description of what this would entail.

  • On line 124, there is mention of a need to reduce the number of CSOs. There needs to be some basis of comparison-how many CSOs are there in a typical year (or other time period)?

The remainder of the comments (below) are largely editorial in nature. It almost appears as if different individuals composed different sections of the manuscript.  It is strongly suggested that the authors closely proof the rest of the manuscript (after Table 1) for word choice and spelling (especially plural usage).

Line 42: severer is a word, but severe (or “the most severe”) would make more sense. Since severer can only appropriately be used to compare two cases, indicating that one is worse than the other.

Line 50: replace “scare” with scarce

Line 62: knowledge of rather than knowledge on

Line 65: High benefit-cost ratios—what does this mean? Greater than 1 (positive)? Something else? Context is needed

Line 73: Undoubtedly there is the need to gain further insight, but why/how/in what context(s)? Is it solely the need to study specific geographic locations or are there other needs/factors that should be explored? What are the holes in the existing literature that this manuscript is addressing?

Line 84: replace “tab” with tap

Line 89: extra space between of and Oslo

Line 102: the word meter should be plural (meters)

Line 105: the word month should be plural (months)

Lines 111-112: the meaning is understandable; the phrasing is not optimal. Suggested modification: “the current combined system has insufficient capacity to address severe rainfall events, which leads to urban flooding and damage to buildings and other infrastructure”

Table 1 (and every situation in which the word Wadi is used)-the plural of wadi is, presumably, wadis. Wadi’s implies a possessive use of the term wadi, which makes no sense.
